# OsNAC15 Regulates Tolerance to Zinc Deficiency and Cadmium by Binding to *OsZIP7* and *OsZIP10* in Rice

**DOI:** 10.3390/ijms231911771

**Published:** 2022-10-04

**Authors:** Junhui Zhan, Wenli Zou, Shuangyuyan Li, Jichun Tang, Xiang Lu, Lijun Meng, Guoyou Ye

**Affiliations:** 1CAAS-IRRI Joint Laboratory for Genomics-Assisted Germplasm Enhancement, Agricultural Genomics Institute at Shenzhen, Chinese Academy of Agricultural Sciences, Shenzhen 518120, China; 2School of Agriculture, Jiangxi Agricultural University, Nanchang 330045, China; 3Rice Breeding Innovations Platform, International Rice Research Institute (IRRI), Metro Manila 1301, Philippines

**Keywords:** zinc deficiency, cadmium tolerance, *OsNAC15*, rice, *OsZIP7*, *OsZIP10*

## Abstract

Zinc (Zn) deficiency and cadmium (Cd) stress are severe threats to the growth and development of plants. Increasing Zn content and/or decreasing Cd content in grain are also important objectives of rice breeding. However, the molecular mechanisms of Zn deficiency tolerance (ZDT) and Cd stress tolerance (CDT) are largely unknown in rice. Here, we report that a NAM/CUC2-like transcription factor, *OsNAC15*, contributes to ZDT and CDT in rice. Knockout of *OsNAC15* reduced ZDT and CDT at the vegetative stage. *OsNAC15* expresses in all tissues of different developmental stages, and is repressed by Zn deficiency and induced by Cd stress. OsNAC15 is a functional transcription factor with transactivation and DNA binding activities. Expression analysis of rice ZIP family genes suggested that the knockout of *OsNAC15* activates or inhibits their transcriptions under Zn deficiency or Cd stress conditions. The yeast one-hybrid assay, transient transcriptional activity assay using the dual-luciferase reporter system and electrophoretic mobility shift assay demonstrated that OsNAC15 directly binds to the zinc deficiency-responsive element motifs in the promoters of *OsZIP7* and *OsZIP10* to repress their transcriptions. The *OsNAC15–OsZIP7/10* module is an essential foundation for further study on the regulatory mechanisms of ZDT and CDT in rice.

## 1. Introduction

Zinc (Zn) is an essential micronutrient for plant, animals and human growth and development, and plays catalytic and structural roles in many proteins, including enzymes [1,2]. “Hidden hunger” is often due to micronutrient deficiency, such as iron (Fe) and Zn deficiency [3]. As one of the most commonly deficient minerals, Zn deficiency restricts crop nutritional quality and yield, raising the risk of Zn deficiency in humans who mainly obtain Zn from crops [4,5,6]. Cadmium (Cd) is a non-essential toxic heavy metal, and the accumulation of Cd can lead to severe damage to plants such as inhibition of growth, cell membrane damage and imbalance of nutrient homeostasis [7,8]. More importantly, Cd accumulation severely threatens human health through the food chain [9]. An adequate concentration of Zn and a tolerable concentration of Cd are fundamental in rice, since rice is the staple food of more than half of the world’s population, particularly in developing countries. Increasing rice grain Zn content through breeding (conventional and transgenic approaches) has been a primary objective of rice biofortification programs. Similarly, reducing grain Cd content is critically important in many rice-producing areas such as China and Japan. Although the correlation between ZDT (CDT) and grain Zn (Cd) content is not well understood, simultaneous improvement tolerance and grain accumulation are desired. Therefore, a better understanding of the molecular mechanisms underlying Zn hemostasis and responses to Cd stress will help to improve breeding efficiency.

Since Zn and Cd have many similar characteristics in physical and chemical aspects, they compete with each other for transporters [10,11,12,13]. Many ion transporters, including ZIP proteins (ZRT-IRT-related protein) and HMA proteins (heavy metal ATPases), are involved in the uptake, sequestration and translocation of Zn and Cd in rice [14,15,16]. OsZIP1 is an efflux transporter that reduces rice’s Zn and Cd accumulation [17]. OsZIP7 plays a critical role in xylem loading to deliver Zn and Cd to developing tissues and grain in rice. The knockout of *OsZIP7* increases Zn and Cd accumulation in roots and basal nodes, and reduces the translocation from root to shoot [15]. OsZIP5 and OsZIP9 are necessary influx transporters and function synergistically in Zn and Cd uptake in rice roots, and the overexpression of *OsZIP9* significantly increases Zn and Cd accumulation in the above-ground tissues and brown rice [18,19,20]. OsHMA3 mainly functions in the vacuoles in rice roots [21]. The loss-of-function allele of *OsHMA3* increases Cd accumulation in the shoots and grain [16], while the overexpression of *OsHMA3* reduces Cd accumulation in the grain and enhances CDT and the expression of genes associated with Zn uptake and translocation [14,22]. Similar to OsZIP7, OsHMA2 also functions in Zn and Cd xylem loading and translocation from root to shoot. The suppression of *OsHMA2* increases Zn concentration in roots but reduces Zn and Cd concentrations in grain [23]. OsZIP10 is a functional Zn transporter in yeast [20] and is associated with rice grain Zn content [24]. The loss-of-function of *OsZIP10* decreases grain Zn and Fe concentration in rice [25]. OsIRT1 is sensitive to excess Zn and Cd and increases Zn and Cd accumulation in roots and shoots and Zn content in grain [26]. Therefore, it is necessary to understand the roles of essential genes in regulating Zn and Cd tolerance and accumulation.

The NAC (NAM, ATAF, and CUC) transcription factors (TFs) comprise a large, plant-specific family. In rice, 151 NAC TFs have been identified, many of which are reported to play essential roles in various biological processes, including biotic and abiotic stress [27,28]. OsNAC2 (SNAC1), OsNAC048 (SNAC2), OsNAC3, OsNAC5, OsNAC6, OsNAC9, OsNAC10, OsNAC14, OsNAC016, OsNAC022 and OsNAC45 are involved in drought and/or salt stresses [29,30,31,32,33,34,35,36]. ONAC127 and ONAC129 synergistically regulate heat stress response and grain filling by coordinating multiple pathways at reproductive stages [37]. OsNTL3 (OsNAC8) regulates heat stress via activating *OsbZIP74* transcription, and the OsNTL3–OsbZIP74 circuit module functions in communications among the endoplasmic reticulum, plasma membrane and nucleus under heat stress [38]. OsNAC111, ONAC122 and ONAC131 are involved in defense against rice blast [39,40]. In addition, many NAC TFs are involved in plant growth and development. OsNAC016 interacts with OsGSK2 and OsSAPK8 to regulate brassinosteroid (BR)-mediated plant architecture [41]. OsCUC1 (ONAC092) and OsCUC3 (ONAC107) function redundantly in regulating rice meristem/organ boundary specification, and *OsCUC1* is negatively regulated by OsmiR164c [42]. Overexpression of OsmiR164b-resistant *OsNAC2* changes plant architecture and improves rice yield [43]. OsNAC2 is also reported to regulate leaf senescence via ABA biosynthesis in rice [44]. NAC TFs are also found to participate in response to nutritional stress. Overexpression of *OsNAC68* improves rice nitrogen use efficiency and grain yield under low nitrogen conditions [45]. *OsNAC4*, *OsNAC5* and *OsNAC6* are induced under Fe toxicity stress in the plant roots [46]. Under aluminum treatment, 25 NAC TFs exhibited different expression patterns among rice cultivars [47].

For the regulation of CDT, Tan et al. [48] reported that five rice NAC TFs were among the Cd-regulated differentially expressed genes (DEGs), and *OsNAC3* and *SNAC1* were explicitly responsive to different Cd stresses via co-expression network analysis. However, only OsNAC3 and OsNAC300 have been shown to regulate CDT through a transgenic approach. *OsNAC300* is induced by Cd stress, mainly expressed in rice root [49]. The knockout of *OsNAC300* decreases CDT, while its overexpression increases CDT. Wang et al. [50] found that a T-DNA insert mutant of *OsNAC3* had considerably reduced CDT, while the overexpression of *OsNAC3* enhanced CDT in rice. Our previous comparative transcriptome analysis of rice varieties grown under Zn-sufficient and -deficient conditions found that 27 NAC TFs were regulated by Zn deficiency [51]. Among these 27 NAC genes, *ONAC32* (*ONAC056*), *ONAC120*, *ONAC017* (*ONAC030*) and *OsNAC300* were firmly induced by Zn deficiency, implying that NAC TFs may play an important role in response to Zn deficiency. However, to date, none of the rice NAC TFs have been functionally validated for ZDT.

A few other types of TFs have been shown to be involved in CDT. For example, the knockout of *OsMYB45* shows a hypersensitive phenotype under Cd stress conditions, probably due to the reduced expression of catalase (CAT) activity-related genes, including *OsCATA* and *OsCATC* [52]. Overexpression of *TaHsfA4a* enhances CDT in wheat, while the knockdown of *OsHsfA4a* decreases CDT in rice [53]. OsTTA, a rice PHD-finger protein, responds to multiple metals (Cd, Zn, copper (Cu) and manganese (Mn)) and regulates the expression of multiple metal transporters, including *OsNRAMP1*, *OsIRT1* and *OsZIP1*. Mutants of *OsTTA* have lower shoot Cd concentration than WT [54]. Ding et al. [55] reported that overexpression of *OsHB4* increases Cd sensitivity and Cd accumulation in the leaves and grain and suggested that *OsHB4* likely regulates the expression of *OsHMA2* and *OsHMA3*.

In a previous study, Calayugan et al. [56] identified the QTL *qZn7.2* with a peak SNP at 29.35 Mb of chromosome 7 for grain Zn concentration. A NAC transcription factor, *OsNAC15* (LOC_Os07g48550/Os07g0684800) is among the annotated genes in this QTL region (300 kb from the peak SNP). In the present study, we demonstrated that OsNAC15 is a functional transcription factor and plays critical roles in ZDT and CDT. OsNAC15 locates in the nucleus, is repressed by Zn deficiency and is induced by Cd stress. Knockout of *OsNAC15* decreases ZDT and CDT at the vegetative stage. Knockout of *OsNAC15* affects many rice ZIP family genes under Zn deficiency or Cd stress. Further analysis demonstrated that OsNAC15 directly binds to the promoters of *OsZIP7* and *OsZIP10* and inhibits their transcriptions.

## 2. Results

### 2.1. Sequence Analysis of OsNAC15

The full-length open reading frame of *OsNAC15* was cloned from the rice cDNA based on the rice database (https://rapdb.dna.affrc.go.jp/, accessed on 31 March 2022). OsNAC15 consists of three exons and two introns encoding a 301-amino-acid protein with a predicted molecular weight of 34.1 kDa. OsNAC15 belongs to the subgroup I–4, which contains 11 members; we used these proteins to construct the phylogenetic tree. Phylogenetic analysis showed that OsNAC15 is closely related to ONAC039, ONAC045, OsNAC300 and ONAC123, showing 49.05%, 39.06%, 36.93% and 25.56% identity with amino acids, respectively (Appendix A). Among these TFs, OsNAC300 has been shown to be involved in CDT, and its expression is also induced by Zn deficiency [49,51]. We further conducted phylogenetic analysis in other species and showed that OsNAC15 is closely related to a hypothetical protein, OsJ_25622, sharing 93.36% identity (Appendix A). In addition, NAC079_1 (*Zea mays*), MINAC11 (*Miscanthus lutarioriparius*) and TaNAC020_2D.2 (*Triticum aestivum*) shared 55.11%, 54.93% and 54.42% identity with OsNAC15, respectively (Appendix A). MINAC11 has been shown to involve various abiotic stresses (salt, drought, cold, ABA, MeJA and SA) [57]. These results suggest that OsNAC15 and its homologs in other species may have some similarities in physiological and biochemical functions. Furthermore, we used MEGA11 to align protein sequences of OsNAC15 and its homologs. The OsNAC15 protein and its homologs contain a conserved N-terminal NAM domain (Appendix A).

### 2.2. Expression Patterns of OsNAC15

The expression profiles of *OsNAC15* were analyzed using RT-qPCR. Under hydroponic culture conditions, expression levels of *OsNAC15* were higher in roots than in shoots and basal stems (Figure 1A,B). Under Zn deficiency conditions (0.04 µM), the expression levels of *OsNAC15* in basal stems and shoots were significantly decreased, but not altered in roots compared with the control condition (0.4 µM) (Figure 1A). Under Zn toxicity conditions (4 µM), the expression levels of *OsNAC15* in root, basal stems and shoots were all significantly decreased compared with the control condition (Figure 1A). The expression of *OsNAC15* was strongly induced by Cd and deficiency of Fe, Mn or Cu, and the strongest induction was observed for Cd (Figure 1B). Under the normal field growth conditions, *OsNAC15* expression was detected in all tissues tested at the tilling, flowering and grain filling stages (Figure 1C).

### 2.3. Subcellular Localization of OsNAC15

To detect the subcellular localization of OsNAC15, the tobacco epidermal cells and the roots of transgenic rice seedlings fused with OsNAC15-GFP were used. In the tobacco epidermal cells expressing GFP alone, the GFP signal was observed in the nucleus and cell membrane (Figure 2). The GFP signal was observed in the nucleus when the OsNAC15-GFP was transiently transformed into the epidermal cells of rice roots. Consistent with the observation of tobacco epidermal cells, the GFP signal in transgenic seedling roots was also detected in the nucleus. These results suggest that OsNAC15 is localized in the nucleus.

### 2.4. Knockout of OsNAC15 Reduces Zn and Cd Tolerance at the Vegetative Stage

To investigate the role of *OsNAC15* in rice, we generated *OsNAC15* knockout lines by the CRISPR/Cas9 editing system. Two independent lines, *osnac15-1* with 5 bp deletion (CAACC deletion) and *osnac15-2* with 1 bp insert (T insert) at the first exon (Appendix A), were used in all experiments. These inserts or deletions result in early transcoding mutations (Appendix A).

We grew the WT and two independent knockout lines in a 1/2 Kimura B solution containing 0.04 or 0.4 µM Zn for 28 days. Under 0.4 µM Zn (control), the lengths of roots and shoots in *osnac15-2* were significantly shorter than in WT (Figure 3A). Under 0.04 µM Zn (deficiency), the root lengths of the two knockout lines were significantly shorter than in WT, while only the shoot length of *osnac15-1* was obviously shorter than that of WT (Figure 3A). However, the root dry weight and shoot dry weight did not differ between different lines under both control and deficiency conditions (Figure 3B). Since a significant difference was found between the WT and knockout lines under the control condition, we evaluated ZDT using the relative trait values calculated as 100× deficiency/control. Although formal statistical tests cannot be conducted, the relative trait performance is a better indicator than the absolute trait value when lines show obvious differences under control. For root length, similar relative values were observed for the knockout lines and WT (Figure 3D). For shoot length, a much greater reduction was observed for the knockout lines than for the WT (Figure 3D). For root dry weight, under Zn deficiency conditions, the knockout lines were 84.30% and 87.51% of their values under control, while the WT was 93.87% of its value under control (Figure 3E). Similarly, for shoot dry weight, the knockout lines showed much lower relative values than the WT (Figure 3E). These results indicate that the knockout of *OsNAC15* reduces ZDT in rice at the vegetative stage.

We then compared the Zn concentration in roots and shoots of the WT and knockout lines under control and deficiency conditions. Although no significant differences were found in the Zn concentration of the roots and shoots between the WT and the knockout lines (Figure 3C), the relative Zn concentration of the knockout lines was obviously lower than that of the WT (Figure 3F). These results suggest that the knockout of *OsNAC15* decreases the Zn accumulation in roots and shoots.

When the WT and knockout lines were grown under 20 µM Cd for 18 days, the root length and shoot dry weight were significantly lower in the knockout lines than in the WT, while the shoot length and root dry weight did not differ between different lines (Figure 4A,B). Similar to ZDT, we also evaluated CDT using the relative trait values, calculated as 100×Cd_treatment/control. For root length and shoot length, the knockout lines were lower than WT (Figure 4D). The knockout lines had lower relative values for root dry weight (24.26%, 26.61%) than the WT (32.91%), and lower shoot dry weight (31.20%, 30.50%) than the WT (41.08%) (Figure 4E). We also compared the Cd concentration and root-to-shoot translocation in the WT and knockout lines. Although the Cd concentration did not differ between WT and the knockout lines (Figure 4C), the root-to-shoot translocations of the knockout lines were significantly increased compared to the WT (Figure 4F). These results indicate that knockout of *OsNAC15* decreases CDT and increases root-to-shoot Cd translocation in rice.

### 2.5. OsNAC15 Is a Functional Transcription Factor

As for most of NAC TFs, OsNAC15 has two important functional domains: conserved NAC domains at the N-terminal, which determine DNA binding activity, and variable domains at the C-terminal, which determine transactivation activity. To determine whether OsNAC15 has transactivation activity, we constructed a series of truncations of OsNAC15, including the N-terminal region, the C-terminal region and various truncated C-terminal regions. Yeast cells transformed with full-length amino acids (1–301 aa), C2-terminal amino acids (172–241 aa) and C3-terminal amino acids (172–201 aa) grew well on the SD-Trp/-His medium, while C-terminal amino acids (172–301 aa) grew weakly (Figure 5A). However, yeast cells transformed with N-terminal amino acids (1–171 aa), C1-terminal amino acids (172–271 aa) and empty BD vector did not grow on the SD-Trp/-His medium. These results indicate that OsNAC15 has transactivation activity, and the activation site locates in the 172–241 amino acids of the C-terminal, and that an unknown domain that inhibits transcriptional activation may exist in the 241–271 amino acids of the C-terminal. Many genes important for zinc deficiency response contain the characteristic cis zinc deficiency response elements (ZDRE), “RTGTCGACAY”, and the ZDRE motif was conserved in higher plants [58]. We performed the yeast one-hybrid assay to test whether OsNAC15 can bind to the ZDRE motif. OsNAC15 was able to bind the three tandem copies of the ZDRE motif (Figure 5B). Taken together, OsNAC15 is a functional transcription factor with both transactivation activity and DNA binding activity.

### 2.6. Knockout of OsNAC15 Changes Expression of Rice ZIP Family Genes

In our yeast one-hybrid assay, OsNAC15 can bind to the ZDRE motif (Figure 5B). The ZIP family proteins are important transporters involved in the absorption, transport and distribution of metals in rice [15,17,19,59,60,61,62]. We analyzed the expression of all 16 rice ZIP family genes in the WT and *osnac15* mutant lines using RT-qPCR. All the ZIP genes were responsive to Zn deficiency and Cd stress. Under the control condition, *OsZIP9* was upregulated while *OsZIP2*, *OsZIP3*, *OsZIP4*, *OsZIP5*, *OsZIP6*, *OsZIP8*, *OsZIP11*, *OsZIP13*, *OsZIP14*, *OsZIP16*, *OsIRT1* and *OsIRT2* were downregulated in the *osnac15* mutants compared to the WT (Figure 6). Under Zn deficiency, *OsZIP1* and *OsZIP16* were downregulated and the other genes were upregulated in the *osnac15* mutants compared to the WT. Under Cd stress, *OsZIP16* was downregulated and the other genes were upregulated in the *osnac15* mutants compared to the WT. These results suggest that OsNAC15 regulates rice ZIP family genes in response to ZDT and CDT.

### 2.7. OsNAC15 Negatively Regulates OsZIP7 and OsZIP10 Transcriptions through Binding to the ZDRE Motif in Rice

To investigate whether OsNAC15 can directly bind to the promoters of ZIP transporters in rice, we first analyzed the ZDRE motif in the promoters of all 16 ZIP transporters. The promoters of *OsZIP5*, *OsZIP7*, *OsZIP9* and *OsZIP10* were found to harbor the typical ZDRE motif (Figure 7A). Then, we performed the yeast one-hybrid assay to test whether OsNAC15 binds to the promoters of these four ZIP genes. Yeast cells co-transformed with pGADT7-Rec2-*OsNAC15* and pHIS2-*OsZIP7pro* or pHIS2-*OsZIP10pro* grew normally on the SD-Leu/-Trp/-His medium with 100 mM 3-AT, but yeast cells co-transformed with pGADT7-Rec2-*OsNAC15* and pHIS2-*OsZIP5pro* or pHIS2-*OsZIP9pro* failed to grow (Figure 7B), indicating that OsNAC15 directly binds to the promoters of *OsZIP7* and *OsZIP10*, but not to the promoters of *OsZIP5* and *OsZIP9*. To further locate the binding motif, the promoters of *OsZIP7* and *OsZIP10* were divided into five fragments. The fragments of *OsZIP7-4* and *OsZIP10-3* were found to be bound by OsNAC15, suggesting that OsNAC15 binds to the ZDRE motif in the promoters of *OsZIP7* and *OsZIP10* (Figure 8A,B). Unexpectedly, the fragments of *OsZIP7-3*, *OsZIP7-5*, *OsZIP10-4* and *OsZIP10-5* were also bound by OsNAC15 (Figure 8A,B), implying that there may be other motifs in the promoter fragments that can be bound by OsNAC15.

To further confirm whether OsNAC15 binds to the ZDRE motif in the *OsZIP7* and *OsZIP10* promoters, we performed the transient transcriptional activity assay using the dual-luciferase reporter system. In the dual-luciferase experiment, *OsZIP7pro:LUC* and *OsZIP7-4pro:LUC* or *OsZIP10pro:LUC* and *OsZIP10-3pro:LUC* were co-transiently transformed with empty vector or *35Spro:OsNAC15* into tobacco epidermal cells, respectively. For *OsZIP7*, the LUC:REN ratio in co-infiltrated *OsZIP7pro:LUC* and *35Spro:OsNAC15* was much lower than in co-infiltrated *OsZIP7pro:LUC* and empty vector, and the *OsZIP7-4pro:LUC* and *35Spro:OsNAC15* had a lower LUC:REN ratio than the *OsZIP7-4pro:LUC* and empty vector (Figure 8C). For *OsZIP10*, the *OsZIP10pro:LUC* and *35Spro:OsNAC15* co-infiltration had a lower LUC:REN ratio than the *OsZIP10pro:LUC* and empty vector co-infiltration, and *OsZIP10-3pro:LUC* and *35Spro: OsNAC15* co-infiltration had a lower LUC:REN ratio than *OsZIP10-3pro:LUC* and empty vector co-infiltration (Figure 8C). Therefore, the expression levels of *OsZIP7* and *OsZIP10* were inhibited by OsNAC15 through ZDRE motif binding, which was consistent with the RT-qPCR results that the expression levels of *OsZIP7* and *OsZIP10* were significantly upregulated in the knockout lines compared to the WT (Figure 6). We then conducted an electrophoretic mobility shift assay to test the binding in vitro. GST-OsNAC15 fused protein directly binds the *OsZIP7* and *OsZIP10* biotin probes containing the ZDRE motif, and the protein–DNA complex bands of GST-OsNAC15 and *OsZIP7* or *OsZIP10* cold competitor probes were weaker (Figure 8D). Taken together, these findings demonstrate that OsNAC15 negatively regulates transcriptions of *OsZIP7* and *OsZIP10* through binding to the ZDRE motif.

## 3. Discussion

*OsNAC15* was previously reported to be mainly induced by abiotic stress treatments (cold, drought, salt, submergence, laid-down submergence and ABA) and biotic stress (rice blast) [63,64,65,66]. *OsNAC15* was also strongly induced by As (III) treatment after 12 h and 24 h in shoots of upland rice [67]. We found that *OsNAC15* was repressed by Zn deficiency and toxicity and induced by Cd stress (Figure 1A,B). In addition, the expression of *OsNAC15* was also regulated by deficiency of Fe, Mn and Cu (Figure 1B). Therefore, OsNAC15 may also play an important role in regulating responses to other metals. Previous studies did not test the transactivation and DNA binding activities of OsNAC15. Through the yeast one-hybrid assay, we demonstrated that OsNAC15 has transactivation activity, with the activation site likely being in the 172–241 amino acids of the C-terminal (Figure 5A), and binds the three tandem copies of the ZDRE motif (Figure 5B).

In the present study, knockout of *OsNAC15* showed decreased ZDT and CDT (Figure 3 and Figure 4). OsNAC15 affected Zn and Cd accumulation in roots and shoots and Cd root-to-shoot translocation (Figure 3 and Figure 4). In both roots and shoots, the relative Zn concentration in the *osnac15* mutant lines were obviously lower than those in the WT (Figure 3F). The knockout of *OsNAC15* also greatly increased Cd translocation from root to shoot (Figure 4F), suggesting that OsNAC15 is an important regulator in Zn and Cd uptake and transport. Hu et al. [49] reported that loss-of-function of *OsNAC300* resulted in hypersensitivity to Cd stress, while the overexpression of *OsNAC300* increased CDT. However, no difference was observed in the OsNAC300-OE lines and the wild-type in Cd uptake and transport.

TFs generally modulate downstream target genes by binding to specific motif elements in their promoters to perform specific physiological and biochemical functions. Although OsNAC15 and OsNAC300 are phylogenetically similar [49], they seem to have different DNA binding characteristics. OsNAC300 positively regulated the expression of *OsCHS1*, *OsPR10a* and *OsPR10b* via binding their promoters, while OsNAC15 directly bound to the promoters of ZIP transporters including *OsZIP7* and *OsZIP10* and negatively regulated their transcriptions (Figure 7 and Figure 8). Hu et al. [49] suggested that Cd uptake or transport-related transporters might not be the target genes of OsNAC300. Since they did not actually test the binding ability of OsNAC300 to any Cd transporters, direct binding of Cd transporters by OsNAC300 cannot be ruled out. The promoters of the two target ZIP genes of OsNAC15 have the cis-regulatory element, the CATGTG motif (Appendix A), which was shown to be bonded by OsNAC300 in the promoters of *OsPR10a* and *OsPR10b*, suggesting that OsNAC300 may also bind to *OsZIP7* and *OsZIP10*. In our yeast one-hybrid assay, the promoter fragments containing the CATGTG motif of the *OsZIP7* and *OsZIP10* did not show binding activities, suggesting that the CATGTG motif is not the cis-regulatory motif element for the binding of OsNAC15. Recently, Mohanty [68] reported that NAC TFs, including OsNAC15, may bind to different potential putative cis-elements in the promoters of upregulated genes in tolerant, highly tolerant and extremely tolerant genotypes from diverse backgrounds in response to submergence tolerance.

Previous studies reported that OsZIP7 plays an integral role in the transport of Zn and Cd. The knockout of *OsZIP7* lead to the retention of Zn and Cd concentration in roots and basal nodes and decreased concentration in shoots and seeds of rice [15,69]. As the homolog of *OsZIP7*, the expression of *OsZIP10* was strongly induced by Zn deficiency in rice roots and shoots [70]. The knockout of *OsZIP10* reduced Zn concentration in rice seeds [25]. Combined with these results, we concluded that OsNAC15 mediates ZDT and CDT at least partially through altering Zn and Cd accumulation and translocation by directly binding to the promoters of *OsZIP7* and *OsZIP10*.

Under Zn deficiency conditions, the ZDRE motif in the promoters of ZIP genes were bound by bZIP TFs in rice, wheat, barley and Arabidopsis [58,70,71,72,73]. We found that OsNAC15 also bound to the ZDRE motif in the promoters of *OsZIP7* and *OsZIP10* (Figure 7B and Figure 8A,B). Therefore, the ZDRE motif of ZIP genes can be bound by different types of TFs, highlighting its importance in response to Zn deficiency as an indispensable regulatory pathway. On the other hand, our yeast one-hybrid assay on the truncated *OsZIP7* and *OsZIP10* promoter fragments showed that OsNAC15 binds to fragments *OsZIP7-3*, *OsZIP7-5*, *OsZIP10-4* and *OsZIP10-5* that do not contain the ZDRE motif (Figure 8A,B), implying that other cis-regulatory motif elements in addition to ZDRE exist in the promoters of *OsZIP7* and *OsZIP10* and can be bound by OsNAC15. It has been reported that the cytokine-related transcription factor OsRR22 directly binds to the promoters of *OsZIP1* and *OsZIP5* through cytokinin response elements and activates their transcriptions in rice [74]. Further studies on the relationship between the NAC TFs and cis-regulatory elements in the promoters of ZIP genes will deepen our understanding of the molecular mechanisms of Zn hemostasis.

On the other hand, our yeast one-hybrid assay suggested that OsNAC15 did not bind to the promoters of *OsZIP5* and *OsZIP9* (Figure 7B), although the ZDRE motif exists. Because we did not use other techniques to further investigate the possible binding of *OsZIP5* and *OsZIP9* by OsNAC15, the results can only be regarded as suggestive. Similar to *OsZIP7* and *OsZIP10*, the expression levels of *OsZIP5* and *OsZIP9* were upregulated in the *osnac15* mutants compared with the WT (Figure 6). Previous studies reported that *OsZIP5* and *OsZIP9* are mainly expressed in roots and function synergistically in Zn and Cd uptake [18,19,20]. However, no difference was observed in the expression of *OsNAC15* in roots between the Zn deficiency and normal conditions (Figure 1A). A possible explanation is that the upregulated expression levels of *OsZIP5* and *OsZIP9* in the *osnac15* mutants were caused indirectly by interacting with other genes such as bZIP TFs.

In summary, we found that OsNAC15 regulates ZDT and CDT through binding to the ZDRE motifs in the promoters of *OsZIP7* and *OsZIP10* to inhibit their transcriptions (Figure 9). Further research using double mutants *osnac15oszip7* and *osnac15oszip10* is needed to investigate the roles of the *OsNAC15–OsZIP7/10* module in regulating ZDT and CDT at a phenotypic level. Whether and how OsNAC15 interacts with other TFs to regulate ZDT and CDT remains to be further examined. Importantly, the roles of the *OsNAC15–OsZIP7/10* module in the regulation of Zn and Cd concentrations in grain need to be investigated. In addition, whether OsNAC15, as a TF, directly receives signals of Zn and/or Cd stress from the external environment or needs to receive signal transmission from upstream signal receivers must also be studied further.

## 4. Materials and Methods

### 4.1. Generation of osnac15 Mutants by CRISPR/Cas9 System

The *osnac15* knockout lines were generated in the cv Nipponbare using the VK005-01 vector and performed according to the instructions (Viewsolid, Beijing, China). Briefly, a 20 bp gRNA target sequence (5′-GTCGGTGATCAACCAGCTGG-3′) was designed using the CRISPR-P 2.0 (http://crispr.hzau.edu.cn/CRISPR2/, accessed on 15 May 2021) and specificity detection was conducted using a BLAST search against the rice genome. Two designed target sequences were synthesized and formed the oligoduplex in a 25 µL reaction at 95 °C for 3 min. Then, the oligoduplex was cloned into the linearized vector with BspQI (New England Biolabs (Beijing), Beijing, China) to generate the CRISPR/Cas9 plasmid. The plasmid was transformed into Nipponbare with the Agrobacterium strain EHA105 (Biomed, Beijing, China). To detect the homozygous mutants, genomic DNA was extracted from leaves of transgenic seedlings, and PCR-amplified products were sequenced by the primer (GATGAAGTGGACGGAAGGAAGGAG). The target sequences and PCR amplified primers are shown in Appendix A.

### 4.2. Plant Materials and Growth Conditions

Nipponbare, the wild-type (WT), and *osnac15* knockout lines were used in this study. Seeds were soaked in water in darkness for 2 days. The germinated seeds were placed on the 96-well plates in a 0.5 mM CaCl_2_ solution in a greenhouse at 25–30 °C. Seedlings were grown for 7 days and used for various experiments. For Zn deficiency, the seedlings were transferred to a 1/2 Kimura B solution (pH 5.6) containing 0.04 µM ZnSO_4_. For response to Cd, the seedlings (20 days old) were transferred to a 1/2 Kimura B solution containing 20 µM CdCl_2_. The nutrient solution was renewed every 2 days.

The field experiment was conducted at the experimental farm of the Agricultural Genomics Institute in Shenzhen, Chinese Academy of Agricultural Sciences, Guangdong. Two-week-old seedlings of Nipponbare were transplanted at a spacing of 20 cm between plants in a row and 20 cm between rows. The normal agricultural practices were followed to manage water and protect crops. Samples were taken from five plants per replicate.

### 4.3. Subcellular Localization of OsNAC15

For subcellular localization, the CDS of *OsNAC15* was cloned into the pDONR221 vector using the BP reaction (Invitrogen, Carlsbad, CA, USA) to generate an entry vector, then combined with a pCAMBIA1300-pUbi-GFP-3 × Flag binary vector using the LR reaction to generate the pUbi:OsNAC15-GFP-3 × Flag vector. The construct was transformed into the GV3101 strain (Biomed, Beijing, China) and infiltrated into 3-week-old tobacco leaves by Agrobacterium injection and incubated for 2–3 days. Meanwhile, the roots of the pUbi:OsNAC15-GFP-3 × Flag transgenic seedlings were harvested. The fluorescence of GFP was observed with a confocal laser scanning microscope (LSM 980; Zeiss, Oberkochen, Germany).

### 4.4. Determination of Metal Concentrations in Plant

Shoots and roots from WT and knockout lines were harvested and washed with 5 mM CaCl_2_ three times. All samples were dried at 65 °C for 3–5 days and digested with HNO3 (60% [*w*/*v*]), as described by Sasaki et al. [75]. The concentrations of Zn and Cd were determined using inductively coupled plasma mass spectrometry ICP-MS (7700, Agilent, Palo Alto, CA, USA).

### 4.5. Transcription Activation Assay in Yeast

Transcription activation activity of OsNAC15 was assessed using the Matchmaker Gold Yeast Two-Hybrid System (Clontech, San Jose, CA, USA). The full-length CDS and various truncated sequences of *OsNAC15* were amplified and cloned into the pGBKT7 vector containing the GAL4 DNA binding domain to generate BD-NAC15 (1-301 aa), BD-NAC15N (1–171 aa), BD-NAC15C (172–301 aa), BD-NAC15C1 (172–271 aa), BD-NAC15C2 (172–241 aa) and BD-NAC15C3 (172–201 aa) constructs. The constructs were transformed into the yeast strain AH109 (Huayueyang Biotech Co., Ltd., Beijing, China) and selected on synthetic dropout nutrient medium lacking tryptophan (SD-Trp). Yeast cells were grown at 28 °C for 2–3 days and streaked onto synthetic dropout nutrient medium lacking tryptophan and histidine (SD-Trp/-His) for observation. Primers for the constructs are listed in Appendix A.

### 4.6. Yeast One-Hybrid Assays

Yeast one-hybrid assays were performed using the Matchmaker Gold Yeast One-Hybrid System (Clontech, San Jose, CA, USA). The CDS of *OsNAC15* was amplified and cloned into the pGADT7-Rec2 vector using the SmaI site to generate recombinant plasmid containing the GLA4 activation domain. The promoter fragments of ZIP genes were cloned into the pHIS2 vector digested with EcoRI and MluI sites. The tested promoter fragments include *OsZIP5* (1922 bp, −1976 to −55 bp), *OsZIP7* (2079 bp, −2097 to −19 bp), *OsZIP7-1* (583 bp, −2097 to −1515bp), *OsZIP7-2* (504 bp, −1535 to −1032bp), *OsZIP7-3* (278 bp, −1041 to −764 bp), *OsZIP7-4* (536 bp, −777 to −242 bp), *OsZIP7-5* (244 bp, −262 to −19 bp), *OsZIP9* (2063 bp, −2088 to −26 bp), *OsZIP10* (2217 bp, −2225 to −9 bp), *OsZIP10-1* (629 bp, −2225 to −1597 bp), *OsZIP10-2* (721 bp, −1693 to −973 bp), *OsZIP10-3* (340 bp, −1021 to −682 bp), *OsZIP10-4* (353 bp, −703 to −351 bp) and *OsZIP10-5* (344 bp, −352 to −9 bp). The pGADT7-Rec2-*OsNAC15* was co-transformed with each of the recombinant plasmids of pHIS2 into the yeast strain Y187, and the transformants were selected on synthetic dropout nutrient medium lacking Leu and Trp (SD-Leu/-Trp). Yeast transformants (OD600 = 0.1, 0.01 and 0.001) were streaked onto synthetic dropout nutrient medium lacking Leu, Trp and His (SD-Leu/-Trp/-His) with 100 mM 3-amino-1,2,4-triazole (3-AT) at 28 °C for 2–3 days. The combination of pGADT7-Rec2 and pHIS2 was used as the negative control, and the interaction between OsbZIP46 and *RAB21* was used as the positive control [76]. Primers for the constructs are listed in Appendix A.

### 4.7. Dual-LUC Assays of Transiently Transformed Tobacco Leaves

For the dual-luciferase reporter assays, the pGreenII 0800-LUC and pGreenII 62-SK vectors were used. The CDS of *OsNAC15* was amplified and cloned into the pGreenII 62-SK using XhaI and XhoI sites to generate the effectors. The approximately 2 kb promoter fragments of *OsZIP7* and *OsZIP10* or truncated promoter fragments including *OsZIP7* (536 bp, −777 to −242 bp) and *OsZIP10* (340 bp, −1021 to −682 bp) were cloned into the pGreenII 0800-LUC to generate the reporters. The Renilla luciferase (REN) gene driven by the 35S promoter was used as an internal control. Then, these recombinant plasmids were individually transformed into *A. tumefaciens* strain GV3101 and co-infiltrated into 3-week-old tobacco leaves by Agrobacterium injection and incubated for 2–3 days. The Dual-Luciferase Reporter Assay System kit (Promega, Madison, WI, USA) was used to measure activities of LUC and REN, and the LUC:REN ratios were calculated. Primers for the constructs are listed in Appendix A.

### 4.8. Electrophoretic Mobility Shift Assay

The CDS of *OsNAC15* was cloned into the pGEX-4T1 vector to generate the GST-NAC15 fusion protein construct and transformed into E. coli strain BL21 (Biomed, Beijing, China). The fusion protein was induced with 0.5 mM IPTG at 16 °C for 12 h and purified using BeaverBeads GSH (Beaver, Suzhou, China) according to the manufacturer’s protocols. For the electrophoretic mobility shift assay, the promoter fragments of *OsZIP7* (30 bp, −651 to −622 bp) and *OsZIP10* (30 bp, −724 to −694 bp) containing the ZDRE motif were synthesized as biotin-labeled probes or unlabeled probes (cold competitor probes). Purified GST or GST-NAC15 proteins were incubated with biotin-labeled probes or unlabeled probes in a 20 μL binding reaction at room temperature for 20 min. The reaction products were electrophoresed on an 6.5% polyacrylamide gels in 0.5×TBE buffer for 1 h. The gels after binding reactions were transferred to N+ nylon membranes (Merk Millipore, Darmstadt, Germany), and the labeled probes were detected using a LightShift Chemiluminescent EMSA kit (Thermo Fisher Scientific, Waltham, MA, USA). The electrophoretic mobility shift assay probes used are listed in Appendix A.

### 4.9. RNA Extraction and RT-qPCR

Total RNA was extracted from the various tissues of the *osnac15* mutants and WT using TRIzol reagent (Invitrogen, Carlsbad, CA, USA). Each of tissue was ground with liquid nitrogen using a mortar and pestle and transferred to a 2.0 mL EP tube. Then, 1 mL TRIzol reagent was added to the tube, followed by incubation on ice for 5 min. The mixture was incubated with 200 μL of trichloromethane on ice for 15 min, then centrifuged at 12,000 rpm for 10 min. Next, 500 μL of supernatant was transferred to a 1.5 mL EP tube and mixed with 500 μL of isopropanol, followed by incubation on ice for 15 min and centrifuge at 12,000 rpm for 30 min. The supernatant was removed and 70% ethanol was added to the tube, followed by centrifugation at 12,000 rpm for 10 min. After drying, 50 μL of DEPC water was added to the tube. RNA quality was detected by the Nanodrop and agarose gel electrophoresis. RNA sample was stored at a −80 °C freezer.

The first-strand cDNA was synthesized from 1 μg of total RNA using a HiScript III RT SuperMix kit for qPCR (+gDNA wiper) (Vazyme, Nanjing, China) according to the manufacturer’s instructions. RT-qPCR was performed using a Bio-Rad CFX96 Real-Time PCR Detection System (Bio-Rad, Hercules, CA, USA) and an SYBR qPCR Master Mix (Vazyme, Nanjing, China). The rice *OsActin1* gene was used as an internal reference. Changes in the transcription of each gene were calculated as 2^−ΔCT^ relative to the internal reference. Three biological replicates were performed for each target gene. All primers used for RT-qPCR are listed in Appendix A.

## Figures and Tables

**Figure 1 ijms-23-11771-f001:**
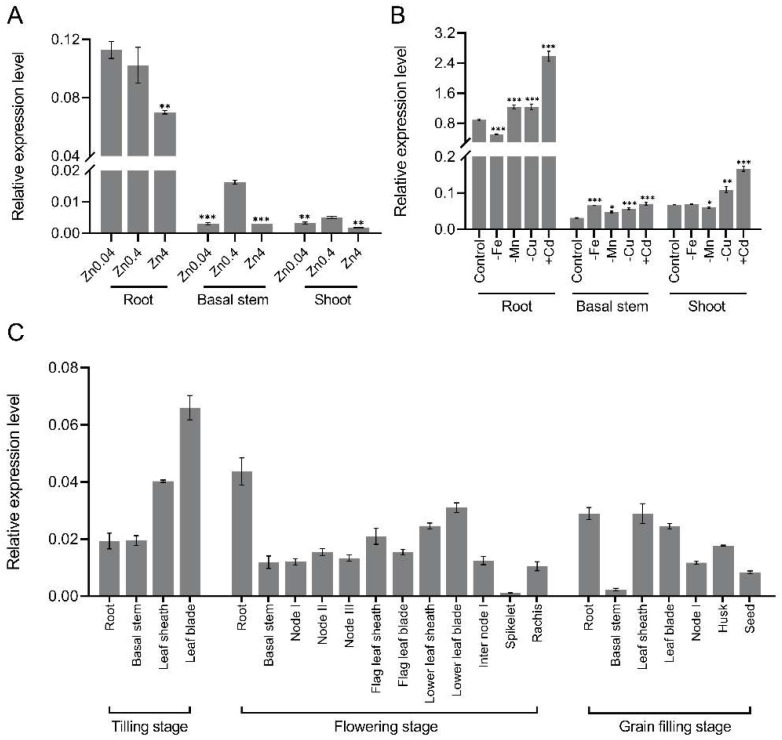
Expression analysis of *OsNAC15*. (**A**) Response of *OsNAC15* expression to different Zn concentrations. Rice seedlings were grown in the 1/2 Kimura B solution containing 0.04 µM, 0.4 µM or 4 µM ZnSO_4_ for 3 days. (**B**) Response of *OsNAC15* expression to Cd stress or metal deficiency. Rice seedlings were grown in the 1/2 Kimura B solution with or without Fe, Mn, Cu or Cd for 3 days. Total RNAs were extracted from the roots, basal stems and shoots of the 13 d Nipponbare seedlings. (**C**) Tissue expression of *OsNAC15* at different growth stages of plants grown in the field. Data are means ± SD of three biological replicates. *, ** and *** indicate significant differences at *p* < 0.05, 0.01 and 0.001, respectively, by Student’s *t*-test.

**Figure 2 ijms-23-11771-f002:**
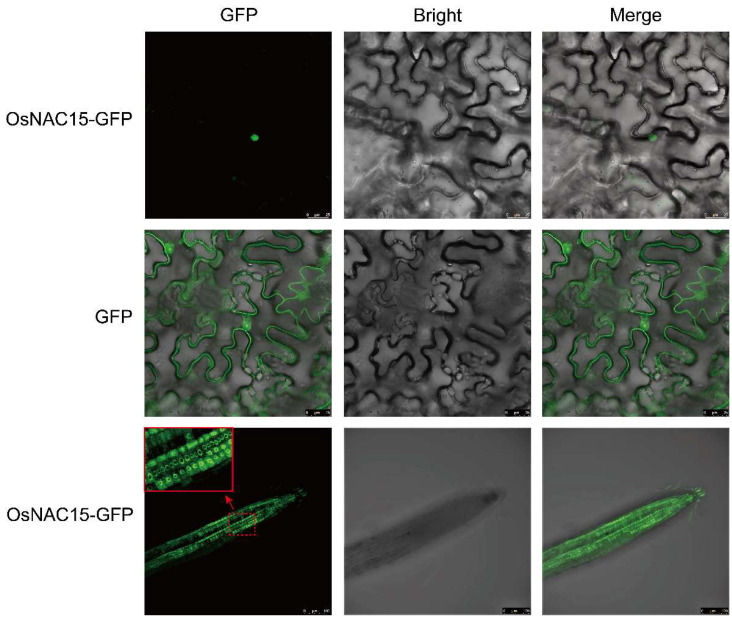
Subcellular localization of OsNAC15. Fusion proteins of OsNAC15-GFP or GFP were expressed in the leaves of tobacco and roots of rice transgenic lines. The image in the red box area is a magnification of the region indicated by the dotted red line. Green fluorescence of GFP was visualized using a laser scanning microscope (LSM 980; Zeiss). Scale bar, 25 µm.

**Figure 3 ijms-23-11771-f003:**
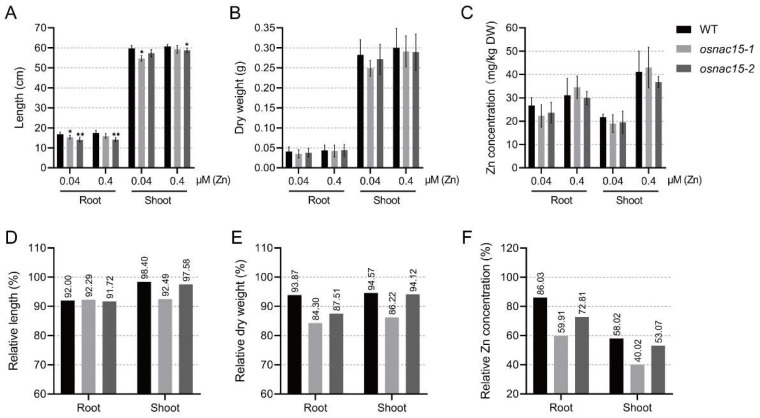
Phenotypic analysis of *OsNAC15* knockout lines for ZDT. Length (**A**), dry weight (**B**), Zn concentration (**C**), relative length (**D**), relative dry weight (**E**) and relative Zn concentration (**F**) of roots and shoots in the WT and knockout lines. The plants were grown in a 1/2 KB solution containing 0.04 μM or 0.4 μM ZnSO_4_ for 28 days. Data are represented as means ± SD of three biological replicates. * and ** indicate significant differences at *p* < 0.05 and 0.01, respectively, by Student’s *t*-test.

**Figure 4 ijms-23-11771-f004:**
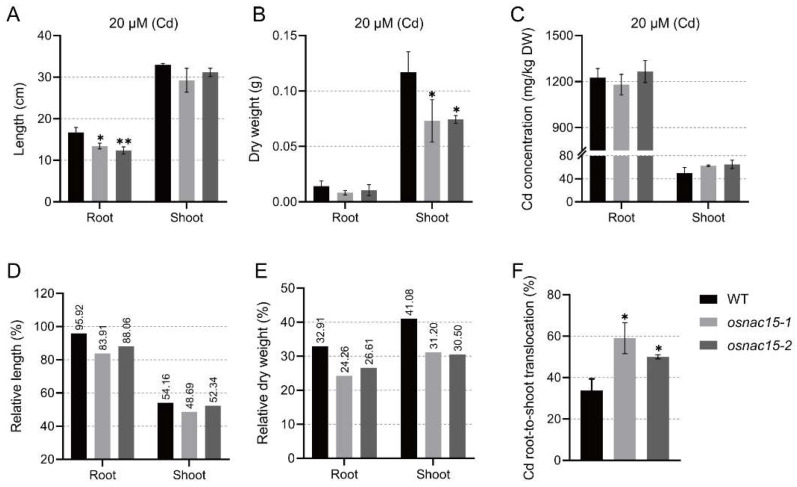
Phenotypic analysis of *OsNAC15* knockout lines for CDT. Length (**A**), dry weight (**B**), Cd concentration (**C**), relative length (**D**), relative dry weight (**E**) of roots and shoots and Cd translocation from root to shoot (**F**) in the WT and knockout lines. The plants were grown in a 1/2 KB solution with or without 20 μM CdCl_2_ for 18 days. Data are represented as means ± SD of three biological replicates. * and ** indicate significant differences at *p* < 0.05 and 0.01, respectively, by Student’s *t*-test.

**Figure 5 ijms-23-11771-f005:**
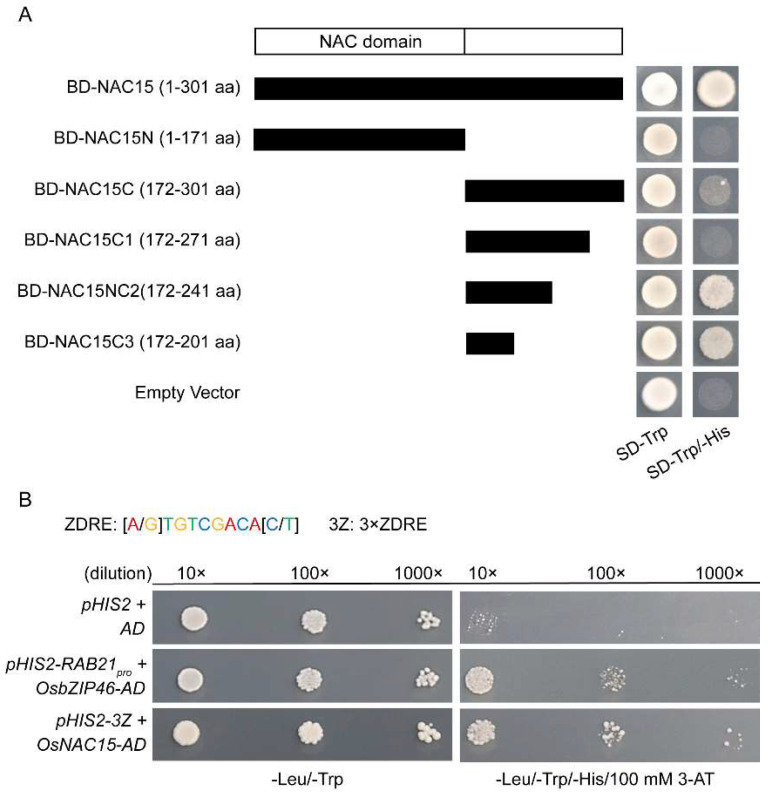
OsNAC15 functions as a transcription factor. (**A**) Transactivation activity of OsNAC15 in yeast. Schematic diagrams of the full-length and truncated OsNAC15 constructs are shown in the left panel. Yeast strain AH109 was used in the transactivation activity analysis. The transformants were streaked on the SD-Trp/-His media (right panel). (**B**) Yeast one-hybrid assay: OsNAC15 binds to the ZDRE motif; 100 mM 3-Amino-1,2,4-triazole (3-AT) was added in the SD-Leu/-Trp/-His medium. Interaction between OsbZIP46 and *RAB21* was used as the positive control. Dilutions of 10×, 100× and 1000× correspond to an OD600 of 0.1, 0.01 and 0.001, respectively.

**Figure 6 ijms-23-11771-f006:**
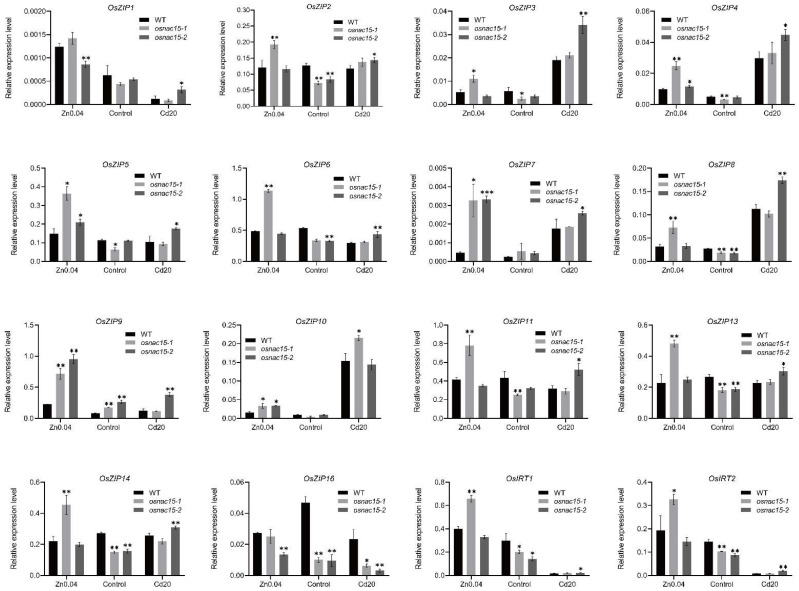
Expression of rice ZIP family genes in the WT and *osnac15* mutants under Zn deficiency and Cd stress conditions. The 10-day-old WT and *osnac15* mutant seedlings were transferred to 1/2 Kimura B solution containing 0.04 µM, 0.4 µM ZnSO_4_ or 20 µM CdCl_2_ for 3 days. Total RNAs were extracted from the roots. Data are means ± SD of three biological replicates. * and ** indicate significant differences at *p* < 0.05 and 0.01, respectively, by Student’s *t*-test.

**Figure 7 ijms-23-11771-f007:**
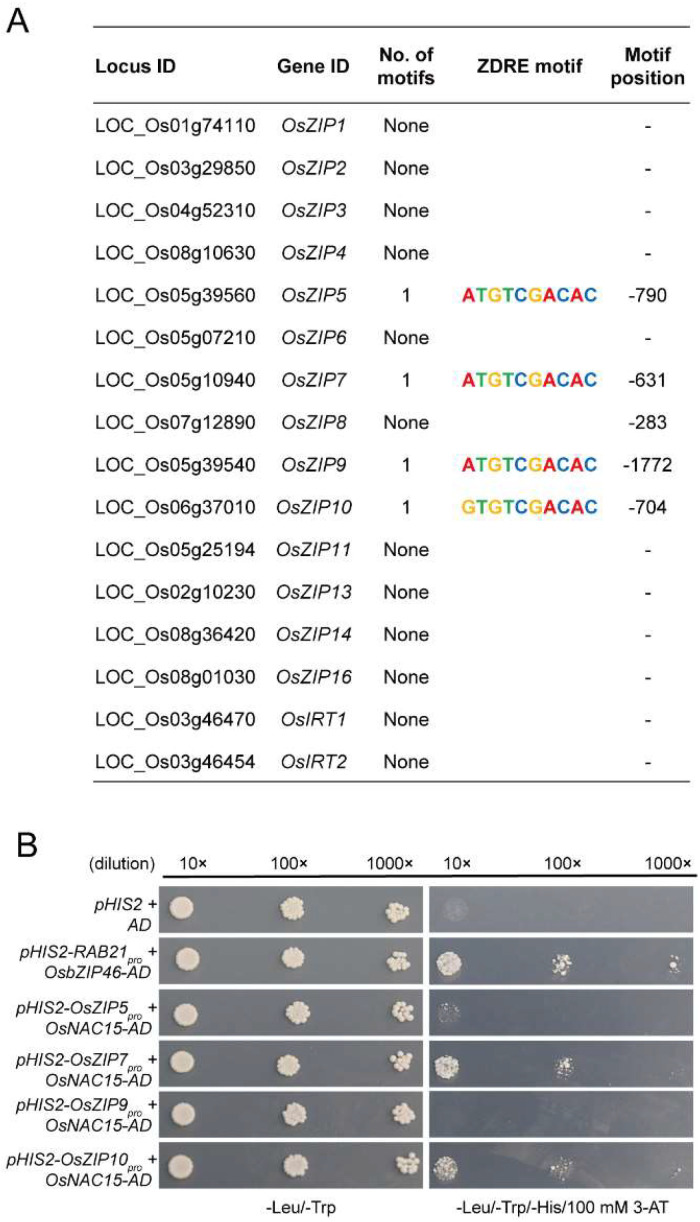
Identification of OsNAC15 target genes among the rice ZIP family genes. (**A**) Analysis of the number and position of detected ZDRE elements in the promoter of each gene. Position of the first nucleotide of the ZDRE motif in relation to the start codon. ZDRE consensus sequence (RTGTCGACAY). (**B**) Yeast one-hybrid assay; 100 mM 3-AT was added in the SD-Leu/-Trp/-His medium. Interaction between OsbZIP46 and *RAB21* was used as positive control. Dilutions of 10×, 100× and 1000× correspond to an OD600 of 0.1, 0.01 and 0.001.

**Figure 8 ijms-23-11771-f008:**
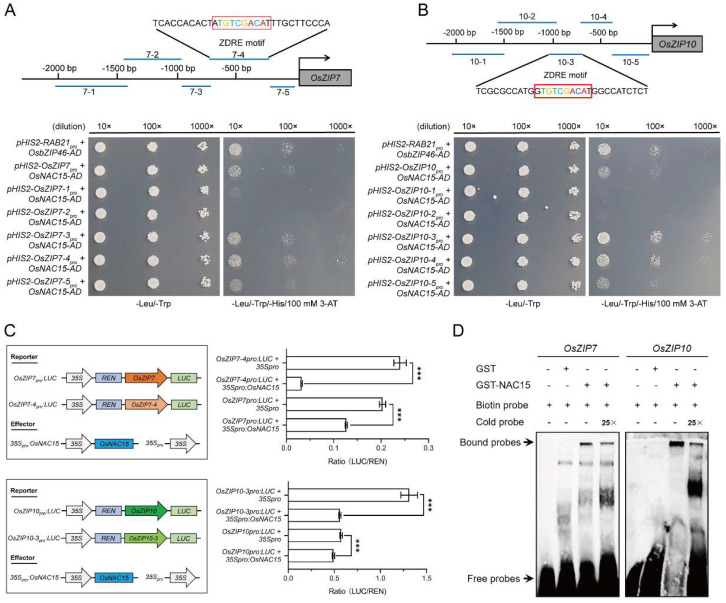
OsNAC15 binds to the promoters of *OsZIP7* and *OsZIP10*. (**A**,**B**) Yeast one-hybrid assay: 100 mM 3-AT was added in the SD-Leu/-Trp/-His medium. Interaction between OsbZIP46 and *RAB21* was used as the positive control. Dilutions of 10×, 100× and 1000× correspond to an OD600 of 0.1, 0.01 and 0.001. (**C**) Transient expression assay in tobacco leaves. Firefly luciferase (LUC) activity was normalized to Renilla luciferase (REN) activity. Schematic representation of various constructs used in the assay is shown in the left panel. (**D**) Electrophoretic mobility shift assay. The competitor probe was added in 25-fold molar excess of labeled probes. Values are means ± SD of three biological replicates. *** *p* < 0.001 by Student’s *t*-test.

**Figure 9 ijms-23-11771-f009:**
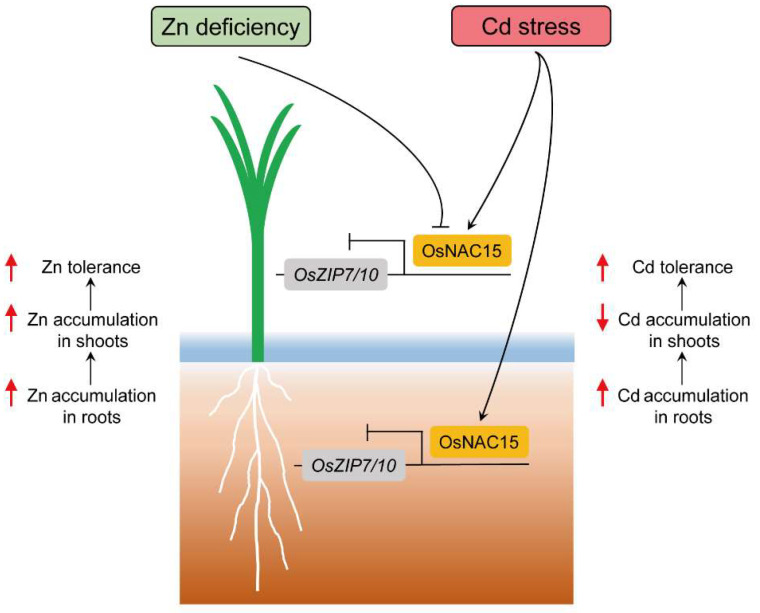
A working model for the OsNAC15-mediated regulation of Zn deficiency and Cd stress tolerance. OsNAC15 directly binds to the *OsZIP7* and *OsZIP10* promoters to inhibit their transcriptions, resulting in increased Zn accumulation and decreased Cd accumulation, thereby enhancing ZDT and CDT.

## Data Availability

Data are contained within the article or Appendix A.

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
