# Peer review of "OsNAC15 Regulates Tolerance to Zinc Deficiency and Cadmium by Binding to OsZIP7 and OsZIP10 in Rice"

_ijms, 2022, doi:10.3390/ijms231911771_

Round 1

Reviewer 1 Report

The manuscript focuses on a very relevant topic of zinc deficiency and cadmium stress tolerance affecting a staple food crop like rice. The manuscript makes an attempt to understand the possible regulatory mechanism involved in coping Zn and Cd stress. 

However, there are certain concerns that must be addressed. 

1. Authors conclude introduction (Line 123) by stating that OsNAC15 binds to promoters of OsZIP7 and 10. They have discussed the role transporter OsZIP7 in introduction. However, there is no mention about OsZIP10 in introduction. Is there no reference for OsZIP10? It would be beneficial for readers to explain the function of OsZIP10 and how it is involved in stress tolerance. 

2. Line 39 - "Grain Zn and Cd contents are especially important in rice." - This line should be rephrased to suggest that maintaining correct concentration of Zn and tolerable concentrations of Cd is important. The current statement is not clear. 

3. Line 127. It would be nice to provide the locus/accession numbers OsNAC15 so that readers can trace the sequence of OsNAC15. Currently only the link to rice database is mentioned in the article. 

4. Line 146 - Is there any specific reasoning as to why the expression of OsNAC15 was only measured under hydroponic conditions.  Is there any data for these under in vitro conditions? 

5. Line 151 - It is important "was only significantly decreased by toxicity, but not altered by deficiency" - This line is confusing. It should be clearly once again stated that authors are discussing about Zn toxicity and Zn deficiency.

6. Line 153 - What are the normal field conditions? What was the size of the field where such trial experiments were done? 

7. Fig 1A and 1B - Why there are two  graphs for roots stacked over each other? Authors should clearly specify the role of each. 

8. Figure 2 - lowermost panel - confocal image on nuclear localisation in roots is of poor quality and out of focus. Image needs to be replaced with a better focus so that GFP localisation could be clearly seen in nucleus. 

9. Unnecessary abbreviations like CK, Y1H should be avoided. It is important to always write the full form of abbreviations in legends for easy understanding of readers.

Author Response

Response to Reviewer 1 Comments

Point 1: Authors conclude introduction (Line 128) by stating that OsNAC15 binds to promoters of OsZIP7 and 10. They have discussed the role transporter OsZIP7 in introduction. However, there is no mention about OsZIP10 in introduction. Is there no reference for OsZIP10? It would be beneficial for readers to explain the function of OsZIP10 and how it is involved in stress tolerance.

Response 1: This is a very important point. We add the details about the function of OsZIP10 in introduction.

Point 2: Line 39 - "Grain Zn and Cd contents are especially important in rice." - This line should be rephrased to suggest that maintaining correct concentration of Zn and tolerable concentrations of Cd is important. The current statement is not clear.

Response 2: Agreed and Actioned.

Point 3: Line 132 - It would be nice to provide the locus/accession numbers OsNAC15 so that readers can trace the sequence of OsNAC15. Currently only the link to rice database is mentioned in the article.

Response 3: Agreed. The locus/accession numbers of OsNAC15 have appeared in Line 122.

Point 4: Line 152 - Is there any specific reasoning as to why the expression of OsNAC15 was only measured under hydroponic conditions. Is there any data for these under in vitro conditions?

Response 4: We used RT-qPCR to check the expression of OsNAC15 under hydroponic conditions. This is a commonly used and acceptable method. We also detected the expression of OsNAC15 in plants grown under the field condition.

Point 5: Line 157 - It is important "was only significantly decreased by toxicity, but not altered by deficiency" - This line is confusing. It should be clearly once again stated that authors are discussing about Zn toxicity and Zn deficiency.

Response 5: This is a very important point. We described the expression levels of OsNAC15 separately under Zn deficiency and Zn toxicity.

Point 6: Line 164 - What are the normal field conditions? What was the size of the field where such trial experiments were done?

Response 6: Thanks for these suggestions. We added the details of field experiments in 4.2. Plant Materials and Growth Conditions.

Point 7: Fig 1A and 1B - Why there are two graphs for roots stacked over each other? Authors should clearly specify the role of each.

Response 7: Since the expression levels of OsNAC15 in roots were much higher than those in basal stems and shoots, we divided the Y-coordinate into two parts to better show the differences between treatments in the basal stems and shoots.

Point 8: Figure 2 - lowermost panel - confocal image on nuclear localization in roots is of poor quality and out of focus. Image needs to be replaced with a better focus so that GFP localization could be clearly seen in nucleus.

Response 8: Thanks for pointing out this. (1) We replaced the image high a high quality one. (2) We magnified a region to clearly show nucleus. (3) We added the description of the magnified region to the legend.

Point 9: Unnecessary abbreviations like CK, Y1H should be avoided. It is important to always write the full form of abbreviations in legends for easy understanding of readers.

Response 9: Agreed and Actioned.

Author Response

Response to Reviewer 2 Comments

Point 1: Modifications in the Abstract.

Response 1: Agreed.

Point 2: Modifications in the Introduction.

Response 2: Agreed.

Point 3: Modifications in the Results.

Response 3: Agreed.

Point 3: Line 172- Highlight.

Response 3: We changed the legend of Figure 1 (C).

Point 4: Line 423-We need more details about the subsequent research regarding the role of OsNAC15 regulates ZDT and CDT.

Response 4: Agreed. We added more details.

Point 5: Line 539- Would you please briefly add the RNA extraction protocol?

Response 5: Agreed. We add the details of the protocol RNA Extraction and RT-qPCR in 4.9.